# Effects of an Educational Intervention on Angolan Adolescents’ Knowledge of Human Reproduction: A Quasi-Experimental Study

**DOI:** 10.3390/ijerph16245155

**Published:** 2019-12-17

**Authors:** Natércia Almeida, Andreia Teixeira, José Garcia, Natália Martins, Carla Ramalho

**Affiliations:** 1Department of Gynecology and Obstetrics, Central Hospital of Huambo, Huambo 95, Angola; 2Center for Health Technology and Services Research (CINTESIS), University of Porto, 4200-450 Porto, Portugal; 3Faculty of Medicine, University of Porto, 4200-319 Porto, Portugal; ncmartins@med.up.pt; 4ARC4DigiT, Instituto Politécnico de Viana do Castelo, 4900-347 Viana do Castelo, Portugal; 5Department of Histology and Medical Embryology, University José Eduardo dos Santos, Huambo 95, Angola; jrmolina47@gmail.com; 6Institute for Research and Innovation in Health (i3S), University of Porto, 4200-135 Porto, Portugal; carlaramalho@med.up.pt; 7Department of Obstetrics, Faculty of Medicine, University of Porto, 4200-319 Porto, Portugal; 8Department of Gynecology and Obstetrics, Centro Hospitalar Universitário de São João, 4200–319 Porto, Portugal

**Keywords:** educational intervention, young adolescents, sex education, human biology, sexuality, contraception

## Abstract

*Background and objectives:* Sex education is a necessity and a right of young people in Angola. However, this education is deficient or even absent in various subsystems and, therefore, the impact of an educational intervention on human biology and sexuality was addressed. *Materials and methods:* This quasi-experimental study employed a non-equivalent control group, pre-test post-test design. It was conducted with students from three secondary schools (6th to 12th grade, two public and one private) in Huambo (Angola), between June and December 2017. First, a questionnaire was distributed to assess the students’ knowledge on aspects related to sexual maturation, psychological development, gynecological organs’ anatomy, human fertilization, contraception, and risks of unprotected sexuality. Then, an educational program was developed by the principal investigator along with the school’s moral and civic education and biology teachers selected for a group of students (experimental group, EG); the others constituted the control group (CG). Classes were held on non-working days, on Saturday mornings (8:00 to 10:00 a.m.), so as not to interfere with the school calendar. The initial questionnaire was redistributed two months later to assess the impact of the intervention. *Results:* Of the 589 individuals included (mean age of 16.8 ± 2.5 years), 56.7% were males. EG (n = 241) consisted of students from the public school and CG (n = 348) by students from public and private schools. The last part of the questionnaire consisted of 30 questions to assess students’ knowledge, and in 23 of these questions, both groups showed no differences at baseline. After the intervention, the EG showed significant improvements (*p* < 0.05), while the CG revealed only slight improvements. *Conclusions:* Students from Huambo province have a significant lack of knowledge on human biology and sexuality. Rigorous development and evaluation of interventions addressing multiple individual and environmental level factors is needed, notably for effective education in human biology and sexuality.

## 1. Introduction

Sex education is called the age-appropriate teaching of sex and intimate human relations, that is culturally relevant and provides scientifically correct and realistic information [1,2]. The Human Rights Convention defines children as *“every individual under the age of 18*” and teenage pregnancy as *“any occurring before the age of 18”.* Therefore, pregnancy in this age group is both a cause and a consequence of violating the child rights [3]. The United Nations Population Fund (UNFPA) report pointed that about 20% of girls under the age of 18 were giving birth in developing countries and that 70,000 died from pregnancy—and childbirth-related complications [3]. In 2014, at the time of the first census of general population and housing in Angola, after the civil war, it was found that the country had 25,789,024 inhabitants, mostly females (13,289,983 inhabitants). Of the 18 provinces, Huambo was the fourth most populous, with 2,019,555 inhabitants, of which 65% of the Angolan population was between 0 and 24 years old, with a high birth rate (5.7 children/woman) [4]. In fact, although the United Nations Educational, Scientific and Cultural Organization (UNESCO) has determined that sex education is both a right and a need for youth and adolescents in Angola, this knowledge is deficient or even absent at the various levels of education (primary, secondary, and university students), and its effective implementation has not yet been verified. In addition, it is important to highlight the fact that general education prepares the individual for work, reduces the likelihood of marriage and early pregnancy, as well as the likelihood of acquiring STI, thus optimizing self-esteem and status within the family and community. Obviously, investing in young people, shaping their potential, is also investing in the future of humanity [3]. However, no action has been taken so far to allow the implementation of sex education during a school year, and this delay in passing and implementing legislation will have a negative impact on the goals and empowerment of men and woman as adolescents, young people or adults, reducing birth rates and improving other indicators, such as those associated with the country’s development, especially maternal, peri and neonatal mortality, as well as the number of clandestine abortions and their consequences.

A systematic review published in 2017, which included 200 studies in sub-Saharan African countries, found encouraging results in terms of adolescents’ knowledge and behavior, following the implementation of an educational intervention program on human immunodeficiency virus (HIV) and acquired immunodeficiency syndrome (AIDS) [5]. It is noteworthy that the educational strategies adopted by sub-Saharan African countries aimed not only to promote the assimilation of knowledge/skills related to healthy behaviors’ promotion (e.g., sexual abstinence and procrastination of sex relations onset), but also to reduce the incidence of sexually transmitted infections (STI) [5,6]. In Angola, HIV, hepatitis B (HBV) and C (HCV) virus and syphilis remain the most prevalent STI, with around 166,000 people living with HIV, representing a prevalence of 1.98% in adults between 15 and 49 years. In a study conducted in Luanda in 2003, 4.5% of pregnant women had HIV, 8.1% had antibodies to HBV, and 5.4% were infected with *Treponema pallidum* [7]. More recently, a systematic review and meta-analysis published in 2019 found a higher incidence and prevalence of trichomoniasis and its association with HIV-1 acquisition in sub-Saharan Africa [8]. Moreover, a retrospective analysis performed in Angola between 2005 and 2012 on the use of services to prevent mother-to-child transmission of HIV found an increase from 9 to 347 in the number of health facilities that provides services, and from 12,061 to 314,805 in the number of HIV tests performed in pregnant women. However, despite the advances made, only 46% of HIV+ pregnant women and 36% of children exposed to HIV were receiving antiretroviral prophylaxis (ARV). More interestingly, given the 2018 UNAIDS report in West and Central Africa, only 40% of men and 59% of women are receiving ARV, which are the lowest rates in the country’ sub-regions (Eastern and Southern Africa: 59% men and 73% women). These data reinforce the need to implement other actions for HIV infection prevention in women and children [9].

In this sense, in this pilot study, we intend, in a first phase, to know the students’ knowledge of the 1st (6th to 9th grade) and 2nd cycle (10th to 12th grade) of secondary schools of Huambo city, on sexuality and human reproduction, and then, to implement an educational intervention program in response to the shortcomings detected in the first phase, and, finally, verify if this intervention led to an improvement of the students’ knowledge. In addition, it was also intended to establish a starting point for the future implementation of sex education at secondary level in this region of the country.

## 2. Materials and Methods

A quasi-experimental study, with non-equivalent control group, pre-test and post-test design was conducted with students from three (two public and one private) primary (1st to 5th grade) and secondary (6th to 12th grade) schools of the Huambo (Angola) city, between June and December 2017. We consider this study quasi-experimental because a study manual was designed, elaborated and given to students, containing unknown aspects on anatomy, reproduction physiology and contraception, verified during the first survey, and after that an educational intervention was implemented. Thus, in a first step, a questionnaire was elaborated and considered the opinion of the teachers of some schools, physicians and nurses of gynecology and obstetrics, pediatricians and, finally, the students of the secondary school and first year of the university. A pilot pre-test was applied to a group of 20 students (1st and 2nd cycle of secondary education 6th to 12th grade and 1st year of medicine), selected for convenience to verify the feasibility, clarity, and comprehension of the questionnaire. This questionnaire was structured into sociodemographic and biological variables, and was divided into two parts. The first part has two versions: One version for girls and one for boys, with some differences in gynecological and sexual issues (see Appendix A and Appendix B, respectively), and aimed to collect sociodemographic variables (such as date of birth, education, place of residence, marital status, extracurricular activities, cohabitation with parents, dialogue about sexuality with parents, education, and occupation of parents) and variables related to gynecological and sexual aspects (such as age of menarche, menstrual cycle, duration of menstruation, age of first ejaculation, erection, age of sexual intercourse onset, history of pregnancy and previous abortions, if they have had or have a boyfriend or husband, if they have ever had sex and what kind, to what extent did they achieve the sexual act, aware of contraception and its use and perception of the risks of unprotected sex). The second part was common to both genders (as seen in Appendix C) and mainly contains closed-ended questions in the form of multiple choice, true or false and some open answers, related to sexual maturation and psychological development, identification of gynecological organs’ anatomy, reproductive sexual cells, human fertilization, contraception and risks of unprotected sex, and aimed to gather participants’ knowledge about anatomy and physiology of organs of the sexual and reproductive system. Premarital sex and the disadvantages of early onset of sex life were also covered, as were the risks of early sex, STI, early pregnancy, and clandestine abortions. We do not talk about homosexual relations because of the silence in the country on this subject, although there are already many homosexual individuals. The questionnaires were distributed to the three schools during June 2017. All students from the schools were invited to participate in the study through a statement from the school boards. In this statement, students were informed about the voluntary nature of participation, the purpose, and conditions of the study. All those who agreed to participate and signed the informed consent were included. Each student was assigned with a numeric code to ensure anonymity and to be able to match students answers in this first stage with answers were collected later.

In a second stage, a sex education intervention was performed for students belonging to School 1 (experimental group, EG). Students who did not attend the intervention (Schools 2 and 3) constituted the control group (CG). There was no intervention in the CG, and the statistical purpose was to make an intragroup comparison, each student to himself, so paired, before and after, to see if there were any differences between students with himself before and after, in both the EG and CG. Schools were selected for convenience due to logistic and operational reasons. The intervention design included sessions for dissemination and clarification of doubts, together with the Council of Christian Churches of Angola (CICA), since the majority of population attends the church. It was also necessary to meet with the parents’ commission and with school teachers to give consent to the study, explaining the reasons that motivated the work, interacting to mitigate possible contradictions, clarifying the advantages of the work and decreasing taboos that could negatively influence their execution. These entities were involved because some wanted us to limit the questions to ask and others to raise them. In the educational program, for example, some wanted us to remove condom from the illustrative materials for teaching on contraception, claiming that it could stimulate sexual intercourse. After clarification and discussion between the principal investigator and parents and members of the Council of Christian Churches, it was unanimously accepted to speak openly on all topics.

The intervention program implemented included, in addition to the elaboration of a sex education manual [10], a training program for the course. Sex education manual was developed to support lectures and was the reference material for students. It contains essential aspects of human biological development related to sexuality, psychological development, anatomy and physiology of human reproduction, contraception and risks of unprotected sex. The same was offered to students and a copy was also given to the school library for consultation. Manual preparation was based on a literature review of the subject and data from the hospital report.

The course design involved lectures (with time for questions and answers), group work sessions and individual work. Lectures were given by the principal investigator, accompanied by the school teachers on subjects related to the teaching program: Moral and civic education, biology and the coordinator of extracurricular activities. The inclusion of teachers was mainly due to ethical reasons and also aimed at reaffirming their educational role, as well as avoiding possible doubts, questions and rejection by school authorities, parents, students, and even others social sectors. Lectures were held weekly, each starting with a brief discussion of the content covered in the previous one, to clarify doubts and revise concepts. The days and respective schedules of the course and hours were defined, according to the direction of the school, with students and teachers and researchers involved, taking place on Saturdays, between 8:00 a.m. and 10:00 a.m., avoiding interference with the internal school activities. The lectures began in the first week of September 2017 and were held two lectures per day, each lasting 40 min and with a 20 min interval. Six hours for individual study were added with the help of the sex education manual, and sharing with parents was encouraged to clarify doubts, thus making a total of 14 h. The inclusion of teachers was mainly due to ethical reasons and also aimed at reaffirming their educational role, also avoiding possible doubts, questions and rejection by school authorities, parents, students and even other social sectors.

The third and last stage consisted of applying the second part of the questionnaire (Appendix C), two months after intervention and five months after the first application in both groups, to assess the impact of the intervention on students’ knowledge. In this stage, students who: (1) In the first stage, did not fulfill the personal data; (2) in the second stage, did not attend at least four of the eight lectures of the course; (3) in the last stage, did not enter the code assigned in the first step (not allowing to match the questionnaires), were excluded.

This project was authorized by the Ministry of Education and Health, through its Provincial Offices and other social sectors, as also ethical approval from the Hospital Regional do Huambo (263/GD/09/2013). All students gave their informed consent and were informed that they could withdraw at any time, without any negative repercussions. This study was in accordance with the ethical principles of the Helsinki Declaration.

Data were analyzed using the Statistical Package for Social Sciences (SPSS, IBM Corp., New York, NY, USA) software, version 25.0. Categorical variables were described as absolute and relative frequencies, n (%), while continuous and normally distributed variables were described by mean and standard deviation $\bar{x}\pm SD$ Non-normally distributed continuous variables were described by median and interquartile range, Med [Q1; Q3]. To test the association between two categorical variables, Chi-square or Fisher’s exact tests were used. To compare two paired categorical variables, McNemar test was used. To compare two independent and non-normal continuous variables, Mann–Whitney test was used. To compare two continuous paired and non-normal variables, Wilcoxon test was used. Values of *p* ≤ 0.05 were considered significant.

## 3. Results

Of the 589 students included, 241 (40.9%) constituted the EG and were in the 8th or 9th grade at School 1. The CG was composed of 175 (29.7%) students in the 9th to 12th grade of School 2 and 173 (29.4%) from the 10th to 12th grades of School 3. The comparison between the two groups in relation to sociodemographic data is presented in Table 1.

Table 2 shows the results of the second part of the questionnaire, where the knowledge on the anatomy and physiology of the sexual organs and reproductive system were measured, before and after intervention. Before the intervention, there were no significant differences between groups in most of the 30 questions of the questionnaire. After the intervention, the EG presented 20 questions with significant improvements (*p* < 0.05), whereas the CG presented only significant improvements in six questions.

Regarding the number of correctly identified women’s genitals (P1F), before the intervention, the groups did not significantly differ (*p* = 0.566) and both showed improvements before and after the intervention (*p* < 0.001). Regarding the identification of male genital organs (P1M), the results were similar: Before the intervention, the groups were not significantly different (*p* = 0.066) and both showed significant improvements before and after the intervention (*p* < 0.001).

Regarding the frequency of ovules release (P2DF), there were no differences between groups before the intervention (*p* = 0.778), but after the EG had more correct answers (*p* < 0.001) and improved significantly between the different periods (*p* < 0.001), and the CG worsened (*p* = 0.001).

With regards to the relationship between menarche and reproductive maturity (P3D), there were significant differences between groups in the two evaluation periods (before EG: 30.2% vs. CG: 45.8%, *p* = 0.004; after EG: 57.3% vs. CG: 41.9%, *p* = 0.002). It is noteworthy that, before the intervention, the CG was the one who answered more correctly, while after the intervention, the EG presented the best results. Thus, the EG improved significantly (*p* < 0.001), while the CG showed no significant differences (*p* = 0.542).

Concerning the risks of adolescent sexuality (P4D), there were significant differences between groups before the intervention (*p* = 0.038), where the CG answered more correct questions, but after the intervention, there were no significant differences (*p* = 0.054). Only the EG improved significantly between the two periods (*p* < 0.001). When looking at the relationship between psychological and sexual maturation (P5C), there were no significant differences between groups before (*p* = 0.365), but there were significant differences after the intervention (*p* < 0.001), where the EG revealed a higher percentage of correct answers. In fact, EG significantly improved before and after the intervention (*p* < 0.001). Regarding contraception (P6B), there were no significant differences between groups before (*p* = 0.527), but they existed after the intervention (*p* < 0.001), with the EG having a higher percentage of correct answers. EG improved significantly before and after the intervention (*p* < 0.001).

Table 3 and Figure 1 show contraceptives known to students. Before the intervention, pills and condom were the best-known methods, with no differences between groups (*p* = 0.385 and *p* = 0.527, respectively). After the intervention, the EG showed improved knowledge of all methods (*p* < 0.001), while the CG only improved the knowledge on pills (*p* < 0.001), intrauterine devices (IUD, *p* < 0.001), subcutaneous implant (*p* = 0.028), interrupted intercourse (*p* = 0.013), and total abstinence (*p* = 0.001).

## 4. Discussion

In this study, considering the pre-intervention evaluation, it was found that there is a lack of knowledge on various topics covered at different educational subsystems. Most participants were males, although the country has a larger number of females [4]. In fact, gender asymmetries are common, although in some educational subsystems, especially in higher education, there has already been a slight increase in the number of women in schools compared to men, especially in careers of education and health sectors [11].

The percentage of parents who concluded university and post-graduate education was very high, giving the overall number of individuals holding a university degree in this country. The latest population census revealed that only 2.1% of the population has higher education, and Huambo province ranks the 9th place, with only 1.4% [12]. The results obtained may be due to the students’ lack of knowledge regarding their parents’ level of education. Alternatively, they may represent a niche of the high school population.

Most EG students answered who often talk about sexuality with their parents, while in the CG, most respondents said that they do not usually address this issue with their parents. Indeed, a study in Mozambique revealed that there are major social and family barriers in addressing sexuality issues between adolescents and parents [13], although due to the limited sample size, the results obtained cannot be generalized to the population. In Nigeria, Emelumadu developed a study to assess parents’ perceptions of when to talk to their children about sexuality, and found that most parents said they did not talk to teenage children because they considered it a waste of time, and the reasons were mainly due to lack of parental education [14]. Moreover, considering the percentage of affirmative answers obtained in the EG group, regarding conversations with parents about sexuality, the situation is even more disturbing, and it can be concluded that either family conversations were not effective or parents were not sufficiently educated on what really matters to talk to your children. In fact, if parents are unaware of the basic concepts of sexuality and associated risks, how are children expected to learn from them if they are not taught in schools? Indeed, when the initial questionnaire was applied, many students revealed that they did not know the meaning of various terms, such as erection, ejaculation, menarche, or menstrual period, leaving the question blank or simply asking what it meant.

In addition, half of students answered in the affirmative about the practice of complete (vaginal) sexual intercourse. Kebede, Molla, and Gerensea found that, of those who had sex, 100% had already experienced full vaginal sex, 6.3% had experienced anal, and 5.2% oral sex [15]. In Angola, data provided by the Huambo Central Hospital in the 2013 Annual Report allude to the high rate of unsafe post-abortion curettage, which clearly reveals the practice of penetrative sex. In this hospital between 2010–2013, 7222 curettages were performed, resulting in 17 maternal deaths. In 2012, an unpublished descriptive study conducted at the same hospital found 69.3% of patients aged 13 to 25 years and only 10% used contraceptive methods [10]. More recently, one study found 715 patients undergoing post-abortion curettage and in 84.3% of them the type was undetermined (probably unsafe), with six maternal deaths [16]. The above data reflects what is happening in Huambo, and in the country as a whole, where adolescents begin their sex life at an early age, and why they are not educated in schools or family on how to prevent unwanted pregnancy and how they are unaware of the risks of induce abortion, undergo abortion with improper personnel and under unsafe conditions, and come to hospitals with irreversible complications that often lead to death. These reasons denote the urgency of investing in the education of adolescents on issues related to their sexual and reproductive life. Thus, new generations, rather than using and abusing of penetrative sex, should be encouraged to a broader sex education, more focused on educated decision-making and conscious sexuality.

On the other hand, we found that the age of sexual activity onset is earlier in boys and in the EG. Similar findings were reported by Gebresllasie, Tsadik, and Berhane in Ethiopia, where most of boys (75%) had their first sexual contact before age 18, while only 25% of girls had their first sexual contact before age 18 [17]. Prata et al. [18] in a study of high school students in Luanda province to determine the main predictors of condom use found an average age of 14.4 years in boys and 15.9 in girls. Similarly, Almeida et al. [16] found a higher percentage (40.3%) of first sexual intercourse from 13–15 years old. In South Africa, countless investigations have been conducted on teenage pregnancy, considered a socio-economic and public health problem [17]. Although more emphasis has been placed on girls, in one study, the authors pointed to boys as the main risk factor for teenage pregnancy, and mentioned as one of the reasons the pressure they suffer to match other colleagues or to be accepted in the group [19]. Similarly, Kebede, Molla, and Gerensea, in Ethiopia, reported that, among the 37.9% of students who had the first sexual contact before the age 18, the main motivation for such was in 61.5% for the desire to have their first sexual experience [15], while another study found that 59.8% of boys claimed to be encouraged by their best friend [17]. It is believed that these differences between girls and boys do not correspond to reality and are due solely to the fact that socially and culturally boys tend to refer younger ages to reality, while girls, due to the existing stigma on the subject, choose to answer to ages beyond reality.

In our study, a notorious lack of knowledge on the use of contraceptive methods in both genders was also stated. Indeed, exploratory studies in young African-Americans have shown limited male knowledge on contraceptives, other than condom [20]. In Kenya, an article published in 2018 reported a lack of knowledge about contraceptives use [21]. Another study in Lomé, Togo, including married men, found men’s low involvement in family planning due to cultural beliefs [22]. Additionally, Kebede, Molla, and Gerensea stated that 51.5% of students used contraception inconsistently [15], and our results corroborates the literature data. Many participants did not answer questions about contraceptives, which denotes the ignorance and lack of involvement that boys have about contraceptive methods. Indeed, contraception is still seen in many countries as a problem for women only.

In Uganda, according to Cardoso, the attempt to reduce the level of STIs, considered one of the highest in sub-Saharan Africa, has gone through the adoption of the ABC policy: A “Abstinence”; B “ Befaithfull”, and C “Condom” [18]. This project, which aimed to change the country’s old focus, led to a drastic reduction in HIV-AIDS rates from 30% to approximately 7% [23]. In Angola, a study conducted in 2005 in Luanda with young people (14–24 years), aiming to know the main predictors of gender differences in condom use, revealed a weak association between married/in marital cohabitation and the consistent use of condom. However, a strong association was stated when compared with those living in urban areas, with high school level or those attending school [18]. Indeed, a review article, including 46 studies, stated a strong positive association between individuals with good education and condom use, relatively to those with lower educational level [24]. However, these results do not corroborate those obtained in this study. In fact, contrary to what would be supposed, most students were unmarried, lived in urban areas, and had a high level of education. In addition, they did not use condom. Clearly, the results of this study reflect what happens throughout the country, triggering the high rates of unwanted pregnancies, adolescent births and related complications, such as illegal abortions and maternal-infant morbimortality [10,25,26]. All of this is a strong reason for the urgent need for effective implementation of sex education in Angola.

Regarding students’ knowledge, most questions did not show significant differences between the two groups at the time of the first evaluation. EG improved substantially in paired evaluation. However, there was a noticeable difference between the two groups after the intervention. The improvement observed in the CG may have occurred because they received information from friends, out of curiosity, or simply by chance, while in the EG, the lack of improvement in some questions seems to be due to lack of continuous study, as this content was included in the sex education manual delivered at the beginning of the course. However, the differences were not significant, which reinforces the need for continuing education to change paradigms. In fact, sex education is essential, not only to prevent HIV-AIDS infection and other STI, early pregnancy and motherhood, but also unsafe abortion [3]. A systematic review of the effect of HIV intervention in sub-Saharan Africa showed that most interventions had positive effects, with longer interventions appearing to be more effective [27]. As can be seen, the intervention developed also sought to obtain the most specific information possible on the various areas that affect young people’s sex and gynecological life, to understand the real reasons and factors behind the high number of unwanted pregnancies and abortions. Most of the girls involved have either denied pregnancy or already aborted. However, data provided by hospitals in recent years have revealed that most of abortions seen at the emergency room come from young people living in the host municipality [10]. Negative answers are thought to reinforce the hypothesis that there is a fear of reprisals for banning abortion in the country. Moreover, this prohibition has been contested for several years and justifies the fact that in the vast majority of hospital abortion cases, the source cannot be determined [10,25]. This study also provides important data on parents who could guide the State in adopting better adult and general education policies to address the weaknesses identified and, specifically in the case of parents, the need to train them in themes about sexuality, encouraging them to talk to their children at home, and to help them dispel any doubts about it. This could be a future research topic.

As main limitations of this work, we highlight the subgroups division. The CG consisted of students from the high-polytechnic school, half of whom belonged to public school 1 and the others to private school, to cover public and private education systems. The EG consisted of young people with less education. This division of groups may be considered a limitation, but it was for logistical reasons. On the other hand, we may also have a selection bias, as only the students more interested in sex education took the course (probably those coming from more open families, where these topics are more openly discussed and probably had previous knowledge on this field). Anyway, despite recognizing the weaknesses of the present study, namely the inability to intervene in all students, the short duration of intervention and follow-up, as well as the loss of students after the intervention, promising results were obtained in the short-term, without neglecting the future need for more lasting work. Indeed, changes in human behavior require permanent and multi-ministerial interventions.

## 5. Conclusions

There is a significant lack of knowledge on reproductive organs’ anatomy and functioning, sexuality, and contraception among general education students in Huambo province, with a careless attitude towards unprotected sex. There is a need for rigorous development and evaluation of interventions that address multiple individual and environmental factors, that is, aiming at effective education in human biology and sexuality. However, longer-term studies that also include parents are needed to better assess the medium- and long-term impact, not only on general knowledge, but also on youth and family attitudes towards issues still considered “taboo”, such as sexuality.

## Figures and Tables

**Figure 1 ijerph-16-05155-f001:**
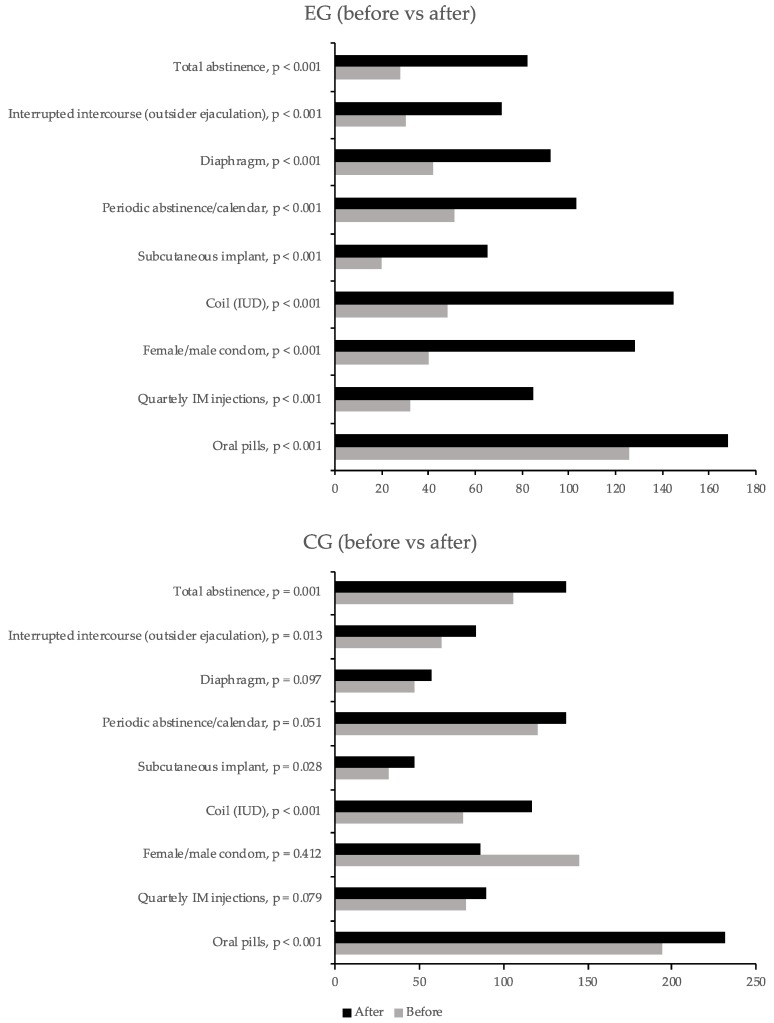
Students’ knowledge on contraceptives before and after the intervention. The *p*-values correspond to the McNemar test.

**Table 1 ijerph-16-05155-t001:** Comparison of students’ knowledge between experimental (EG) and control (CG) groups, at the time of first evaluation, in the primary and secondary schools of Huambo, in June 2017. Source: Own construction.

Variables	EG n = 241	CG n = 348	*p*-Value
**Gender**, n (%)	n = 241	n = 348	**<0.001 ^a,^***
Males	98 (29.3)	236 (70.7)	
Females	143 (56.1)	112 (43.9)	
**Age** (years), Med [Q_1_;Q_3_]	n = 241 15 [14; 16]	n = 348 18 [17; 19]	**<0.001 ^b,^***
**Marital status**, n (%)	n = 230	n = 337	0.086 ^a^
Not married	206 (42.0)	285 (58.0)	
Married or marital^#^	24 (31.6)	52 (68.4)	
**Father’s education**, n (%)	n = 231	n = 304	**<0.001 ^a,^***
No schooling	7 (41.2)	10 (58.8)	
Primary education	1 (11.1)	8 (88.9)	
Secondary education	17 (50.0)	17 (50.0)	
Higher education	20 (23.5)	65 (76.5)	
University education	85 (42.9)	113 (57.1)	
Post-graduation	101 (52.6)	91 (47.4)	
**Mother’s education**, n (%)	n = 234	n = 304	**0.039 ^a,^***
No schooling	12 (44.4)	15 (55.6)	
Primary education	5 (22.7)	17 (77.3)	
Secondary education	17 (38.6)	27 (61.4)	
Higher education	35 (33.7)	69 (66.3)	
University education	101 (48.1)	109 (51.9)	
Post-graduation	64 (48.9)	67 (51.1)	
**Usually talks with parents about sexuality**, n (%)	n = 221	n = 320	**0.008 ^a^**
No	94 (35.2)	173 (64.8)	
Yes	127 (46.4)	147 (53.6)	
**Age of first sexual relation, females** (years), Med [Q_1_; Q_3_]	n = 15 15 [14; 17]	n = 39 16 [16; 17]	**0.017 ^b,^***
**Previous pregnancy, females**, n (%)	n = 137	n = 104	**<0.001 ^a^,***
No	137 (60.1)	91 (39.9)	
Yes	0 (0.0)	13 (100.0)	
**Previous abortion, females**, n (%)	n = 142	n = 108	0.432 ^c^
No	142 (57.0)	107 (43.0)	
Yes	0 (0.0)	1 (100.0)	
**Age of first sexual relation, males** (years), Med [Q_1_;Q_3_]	n = 65 13 [11.5; 14]	n = 187 14 [12; 15]	**<0.001 ^b,^***
**Sexual experience**, n (%)			
**Kisses**	n = 189	n = 307	**<0.001 ^a,^***
No	46 (25.1)	137 (74.9)	
Yes	143 (45.7)	170 (54.3)	
**Caressing**	n = 190	n = 307	0.945 ^a^
No	112 (38.4)	180 (61.6)	
Yes	78 (38.0)	127 (62.0)	
**Masturbation**	n = 189	n = 305	**0.005 ^a,^***
No	166 (41.2)	237 (58.8)	
Yes	23 (25.3)	68 (74.7)	
**Sexual intercourse**	n = 189	n = 309	**<0.001 ^a,^***
No	134 (58.3)	96 (41.7)	
Yes	55 (20.5)	213 (79.5)	
**Age at first sexual intercourse** (years), Med [Q_1_; Q_3_]	n = 72 14 [12; 14]	n = 216 15 [14; 16.75]	**<0.001 ^b,^***
**Feels worried (bad) when engaging in sex**, n (%)	n = 130	n = 254	0.589^a^
No	37 (25.5)	108 (74.5)	
Yes	35 (28.5)	88 (71.5)	

^a^: Chi-square test; ^b^: Mann–Whitney test; ^c^: Exact Fisher test; *: Significant at 5 %; #: Married, those who are officially married by the civil registry; marital, those couples who live together without being officially (legally) married.

**Table 2 ijerph-16-05155-t002:** Results of the 2nd part of the students’ knowledge questionnaire, before and after intervention, by experimental (EG) versus control (CG) groups, in the secondary and high schools of the Huambo, from June to December 2017. Source: Own construction.

Before			After			EG (before *vs.* after)	CG (before *vs.* after)
EG	CG	*p*-Value	EG	CG	*p*-Value	*p*-Value	*p*-Value
**P_1F_-Correct identification of females sexual organs**, Med [Q_1_;Q_3_]:							
n = 241 4 [2.5; 5]	n = 348 4 [0; 5]	0.566 ^a^	n = 210 6 [5; 7]	n = 320 4 [3; 6]	**<0.001 ^a,^***	**<0.001 ^b,^***	**<0.001 ^b,^***
**P_1M_-Correct identification of males sexual organs**, Med [Q_1_;Q_3_]:							
n = 241 2 [1; 4]	n = 348 3 [0; 5]	0.066 ^a^	n = 210 5 [4; 6]	n = 320 4 [2.25; 5]	**<0.001 ^a,^***	**<0.001 ^b,^***	**<0.001 ^b,^***
**P_2AF_-Correct knowledge about denomination of the feminine reproductive cell**, n (%)							
n = 150 123 (82.0)	n = 192 168 (87.5)	0.157 ^c^	n = 201 186 (92.5)	n = 242 208 (86.0)	**0.028 ^c,^***	**<0.001 ^d,^***	1.000 ^d^
**P_2AM_-Correct knowledge about denomination of the masculine reproductive cell**, n (%) n (%)							
n = 154 127 (82.5)	n = 220 190 (86.4)	0.157 ^c^	n = 197 161 (81.7)	n = 247 208 (84.2)	0.488 ^c^	1.000 ^d^	0.742 ^d^
**P_2BF_-Correct knowledge on the production site of feminine reproductive cells**, n (%)							
n = 149 64 (43.0)	n = 195 98 (50.3)	0.179 ^c^	n = 196 124 (63.3)	n = 235 128 (54.5)	0.065 ^c^	**0.001 ^d,^***	**0.016 ^d,^***
**P_2BM_-Correct knowledge on the production site of masculine reproductive cells**, n (%), n (%)							
n = 114 92 (80.7)	n = 196 171 (87.2)	0.121 ^c^	n = 189 155 (82.0)	n = 232 193 (83.2)	0.751 ^c^	0.523 ^d^	0.584 ^d^
**P_2CF_-Correct knowledge about the normal process of female reproductive cell formation**, n (%)							
n = 132 53 (40.2)	n = 162 92 (56.8)	**0.005 ^c,^***	n = 194 94 (48.5)	n = 224 116 (51.8)	0.497 ^c^	0.080 ^d^	0.280 ^d^
**P_2CM_-Correct knowledge about the normal process of male reproductive cell formation**, n (%)							
n = 121 66 (54.5)	n = 180 140 (77.8)	**<0.001 ^c,^***	n = 190 150 (78.9)	n = 224 170 (75.9)	0.460 ^c^	**<0.001 ^d,^***	1.000 ^d^
**P_2DF_-Correct knowledge of the site of reproductive feminine cells release**, n (%)							
n = 110 38 (34.5)	n = 149 54 (36.2)	0.778 ^c^	n = 188 112 (59.6)	n = 207 44 (21.3)	**<0.001 ^c,^***	**<0.001 ^d,^***	**0.001 ^d,^***
**P_2DM_-Correct knowledge of the site of reproductive masculine cells release**, n (%)							
n = 109 69 (63.3)	n = 167 114 (68.3)	0.394 ^c^	n = 182 158 (86.8)	n = 218 158 (72.5)	**<0.001 ^c,^***	**<0.001 ^d,^***	0.486 ^d^
n = 109 69 (63.3)	n = 167 114 (68.3)	0.394 ^c^	n = 182 158 (86.8)	n = 218 158 (72.5)	**<0.001 ^c,^***	**<0.001 ^d,^***	0.486 ^d^
**P_3A_-Answer correctly to the following statement “Menarche is the women’s first period” (True/False)**, n (%)							
n = 159 151 (95.0)	n = 163 155 (95.1)	0.959 ^c^	n = 192 186 (96.9)	n = 219 197 (90.0)	**0.005 ^c,^***	0.581 ^d^	0.227 ^d^
**P_3B_-Answer correctly to the following statement “Menarche’s age is the same in all the women” (True/False)**, n (%)							
n = 157 137 (87.3)	n = 168 147 (87.5)	0.948 ^c^	n = 187 161 (86.1)	n = 213 183 (85.9)	0.959 ^c^	0.678 ^d^	0.581 ^d^
**P_3C_-Answer correctly to the following statement “After the beginning of the menarche the woman can already get pregnant” (True/False)**, n (%)							
n = 163 146 (89.6)	n = 168 154 (91.7)	0.513^c^	n = 189 175 (92.6)	n = 217 189 (87.1)	0.070 ^c^	0.629 ^d^	0.057 ^d^
**P_3D_-Answer correctly to the following statement “After the menarche the woman is already capable of having children” (True/False)**, n (%)							
n = 159 48 (30.2)	n = 166 76 (45.8)	**0.004 ^c,^***	n = 192 110 (57.3)	n = 215 90 (41.9)	**0.002 ^c,^***	**<0.001 ^d,^***	0.542 ^d^
**P_3E_-Answer correctly to the following statement “Is normal that the first menstrual cycles are irregular” (True/False)**, n (%)							
n = 158 142 (89.9)	n = 171 136 (79.5)	**0.010 ^c,^***	n = 191 176 (92.1)	n = 217 178 (82.0)	**0.003 ^c,^***	0.481 ^d^	0.424 ^d^
**P_4A_-Correct knowledge of reproductive organs involved in women’s sexual cycle**, n (%)							
n = 151 46 (30.5)	n = 193 73 (37.8)	0.154 ^c^	n = 189 117 (61.9)	n = 252 88 (34.9)	**<0.001 ^c,^***	**<0.001 ^d,^***	0.532 ^d^
**P_4B_-Correct knowledge about hormones involved in regulating the sexual cycle in women**, n (%)							
n = 85 12 (14.1)	n = 109 29 (26.6)	**0.035 ^c,^***	n = 170 77 (45.3)	n = 153 16 (10.5)	**<0.001 ^c,^***	**<0.001 ^d,^***	**0.031 ^d,^***
**P_4C_-Correct knowledge about when you are most likely to become pregnant**, n (%)							
n = 155 17 (11.0)	n = 194 28 (14.4)	0.337 ^c^	n = 187 49 (26.2)	n = 246 41 (16.7)	**0.015 ^c,^***	**<0.001 ^d,^***	0.472 ^d^
**P_4D_-Correct knowledge about what risk they consider most important when having sex in adolescence**, n (%)							
n = 177 72 (40.7)	n = 260 132 (50.8)	**0.038 ^c,^***	n = 199 120 (60.3)	n = 297 153 (51.5)	0.054 ^c^	**<0.001 ^d,^***	0.918 ^d^
**P_4EA_-Answer correctly to the following statement “The ripe ovule in the woman’s gynecological apparatus stays during every menstrual cycle” (True/False)**, n (%)							
n = 108 30 (27.8)	n = 146 81 (55.5)	**<0.001 ^c,^***	n = 170 85 (50.0)	n = 164 77 (47.0)	0.577 ^c^	**<0.001 ^d,^***	0.216 ^d^
**P_4EB_-Answer correctly to the following statement “The spermatozoids can stay alive in the woman’s genital organs for 10 days” (True/False)**, n (%)							
n = 120 44 (36.7)	n = 167 98 (58.7)	**<0.001 ^c,^***	n = 176 115 (65.3)	n = 187 108 (57.8)	0.138 ^c^	**<0.001 ^d,^***	0.857 ^d^
**P_4EC_-Answer correctly to the following statement “The menstrual cycle is the interval of time that elapses between the first day of period and the first day of the following period” (True/False)**, n (%)							
n = 99 57 (57.6)	n = 145 98 (67.6)	0.111 ^c^	n = 168 129 (76.8)	n = 162 93 (57.4)	**<0.001 ^c,^***	**0.002 ^d,^***	0.289 ^d^
**P_4ED_-Answer correctly to the following statement “A duration of a normal cycle can go from 21 to 35 days. Being like this, the woman can have regular or irregular cycles” (True/False)**, n (%)							
n = 117 89 (76.1)	n = 152 107 (70.4)	0.300 ^c^	n = 176 142 (80.7)	n = 182 126 (69.2)	**0.013 ^c,^***	0.450 ^d^	0.450 ^d^
**P_5A_-Correct knowledge of the most characteristic sign of the masculine puberty**, n (%)							
n = 162 24 (14.8)	n = 234 35 (15.0)	0.969 ^c^	n = 200 106 (53.0)	n = 266 55 (20.7)	**<0.001 ^c,^***	**<0.001 ^d,^***	**0.050 ^d,^***
**P_5B_-Correct knowledge of the most visible characteristic sign of the feminine puberty**, n (%)							
n = 170 125 (73.5)	n = 232 145 (62.5)	**0.020 ^c,^***	n = 202 160 (79.2)	n = 274 188 (68.6)	**0.010 ^c,^***	0.268 ^d^	**0.034 ^d,^***
**P_5C_-Correct knowledge about human sexual maturation**, n (%)							
n = 132 13 (9.8)	n = 198 26 (13.1)	0.365 ^c^	n = 186 80 (43.0)	n = 236 28 (11.9)	**<0.001 ^c,^***	**<0.001 ^d,^***	0.845 ^d^
**P_5D_-Correct knowledge about the basis of sexual relationship onset**, n (%)							
n = 172 72 (41.9)	n = 236 109 (46.2)	0.385 ^c^	n = 200 142 (71.0)	n = 277 116 (41.9)	**<0.001 ^c,^***	**<0.001 ^d,^***	0.476 ^d^
**P_6A_-Correct knowledge about the existing contraceptive methods**, n (%)							
n = 159 98 (61.6)	n = 231 156 (67.5)	0.230 ^c^	n = 196 142 (72.4)	n = 276 185 (67.0)	0.208 ^c^	0.105 ^d^	0.810 ^d^
**P_6B_-Correct knowledge about who should use contraceptive methods**, n (%)							
n = 160 92 (57.5)	n = 239 145 (60.7)	0.527 ^c^	n = 202 164 (81.2)	n = 280 179 (63.9)	**<0.001 ^c,^***	**<0.001 ^d,^***	0.657 ^d^
**P_6C_-Knowledge about the contraceptive methods they had heard about**, Med [Q_1_; Q_3_]							
n = 241 1 [0; 3]	n = 348 1.5 [0; 4]	**0.013 ^a,^***	n = 209 4 [3; 6]	n = 320 3 [1; 5]	**<0.001 ^a,^***	**<0.001 ^b,^***	**<0.001 ^b,^***

^a^: Mann–Whitney test; ^b^: Wilcoxon test; ^c^: Chi-square test; ^d^: McNemar test; *: Significant at 5%.

**Table 3 ijerph-16-05155-t003:** Contraceptives known by the participants, before and after intervention, between experimental (EG) vs. control (CG) groups, in the primary and secondary schools of the Huambo city, from June to December 2017. Source: Own construction.

Questions	Before			After		
	EG	CG	*p*-Value	EG	CG	*p*-Value
**Oral pills**_,_ n (%)	n = 241 126 (52.3)	n = 347 194 (55.9)	0.385 ^a^	n = 209 168 (80.4)	n = 320 232 (72.5)	**0.039 ^a,^***
**Quarterly IM injections**_,_ n (%)	n = 241 32 (13.3)	n = 347 78 (22.5)	**0.005 ^a,^***	n = 209 85 (40.7)	n = 320 90 (28.1)	**0.003 ^a,^***
**Female/Male condom**, n (%)	n = 241 40 (57.5)	n = 239 145 (60.7)	0.527 ^a^	n = 209 128 (61.2)	n = 320 86 (26.9)	**<0.001 ^a,^***
**Coil (IUD)**, n (%)	n = 241 48 (19.9)	n = 347 76 (21.9)	0.562 ^a^	n = 209 145 (69.4)	n = 320 117 (36.6)	**<0.001^a,^***
**Subcutaneous implant**, n (%)	n = 241 20 (8.3)	n = 347 32 (9.2)	0.698 ^a^	n = 209 65 (31.1)	n = 320 47 (14.7)	**<0.001 ^a,^***
**Periodic abstinence/calendar**, n (%)	n = 241 51 (21.2)	n = 347 120 (34.6)	**<0.001 ^a,^***	n = 209 103 (49.3)	n = 320 137 (42.8)	0.144 ^a^
**Diaphragm** n (%)	n = 241 42 (17.4)	n = 347 47 (13.5)	0.196 ^a^	n = 209 92 (44.0)	n = 320 57 (17.8)	**<0.001 ^a,^***
**Interrupted intercourse (outsider ejaculation)**, n (%)	n = 241 30 (12.4)	n = 347 63 (18.2)	0.062 ^a^	n = 209 71 (34.0)	n = 320 84 (26.3)	0.056 ^a^
**Total abstinence** n (%)	n = 241 28 (11.6)	n = 347 106 (30.5)	**<0.001 ^a,^***	n = 209 82 (39.2)	n = 318 137 (43.1)	0.381 ^a^

IUD: Intrauterine device; ^a^: Chi-square test; *: Significant at 5%.

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
