# Peer review of "Effects of an Educational Intervention on Angolan Adolescents’ Knowledge of Human Reproduction: A Quasi-Experimental Study"

_ijerph, 2019, doi:10.3390/ijerph16245155_

Round 1

Reviewer 1 Report

I have checked this revised paper against my initial review comments and am satisfied that most have been addressed. I believe the discussion could be strengthened with reflection on how the findings relate to otherresearch conducted in Angola but the authors have indicated they could not find any. 

Some minor things I noticed:

Line 98 materials and methods – study still referred to as prospective and longitudinal which it is not

Line 146 – what is ‘Condon’? and what is ‘anarchic sexual intercourse?

Line 175 – access should be assess

There remain minor grammatical errors throughout. 

Author Response

I have checked this revised paper against my initial review comments and am satisfied that most have been addressed. I believe the discussion could be strengthened with reflection on how the findings relate to other research conducted in Angola but the authors have indicated they could not find any.

Answer: Thank you for the reviewer comment. Discussion section was further improved.

Some minor things I noticed:

Line 98 materials and methods – study still referred to as prospective and longitudinal which it is not

Answer: We agree with the reviewer comment. The terms prospective and longitudinal were removed.

Line 146 – what is ‘Condon’? and what is ‘anarchic sexual intercourse?

Answer: Both terms were corrected. Condon was replaced by condom and anarchic sexual intercourse by sexual intercourse.

Line 175 – access should be assess

Answer: Thanks. This was a typo; the term was replaced

There remain minor grammatical errors throughout.

Answer: The manuscript was revised accordingly.

Reviewer 2 Report

I would like to thank the authors for the changes done. I have no further comments. 

Author Response

I would like to thank the authors for the changes done. I have no further comments.

Answer: Thank you for the overall appreciation of our work.

Reviewer 3 Report

What kind of sites are being referred to on p. 2, L 83?

I'm not certain why the author said that this is a quasi-experimental design (L 100, p. 3) Were all the students, both E and C given the manual?

I think that The word anarchic on p. 4 L. 147 is incorrect

It looks like the control group consisted of an older population.  The age difference would have an impact on knowledge, as knowledge about human sexuality increases with age and experience.  Is there away to select out the older kids in the control group or compare similar age groups for E & C.

I'm unclear what point the author was trying to make on p, 9 L. 224-226.  It is confusing

p. 10 L. 260 I assume that "address this issue" should  be followed by 'with their parents'

p.10 L. 266-268 doesn't seem to fit.  I think it would help to explain why it is there.

p 10 2nd to last paragraph, L 277-283 is unclear in terms of what point is being made and supported by the literature

There are too many tables in this relatively short paper.  Perhaps some of them can be combined or footnoted.

Author Response

What kind of sites are being referred to on p. 2, L 83?

Answer: Sites refers to health facilities that provides services. This information was included in the revised version of the manuscript.

I'm not certain why the author said that this is a quasi-experimental design (L 100, p. 3) Were all the students, both E and C given the manual?

Answer: Thank you for the reviewer comment. Only the E group received the manual.

I think that The word anarchic on p. 4 L. 147 is incorrect

Answer: This word was removed.

It looks like the control group consisted of an older population. The age difference would have an impact on knowledge, as knowledge about human sexuality increases with age and experience. Is there away to select out the older kids in the control group or compare similar age groups for E & C.

Answer: Thanks for the reviewer comment. Given that E and C groups at baseline had socio-demographic differences, comparisons were made before and after the intervention for better interpretation of results. That is, if at the level of knowledge, there were no differences before the intervention, it is because age do not interfere with the level of knowledge. This could be a limitation if, before the intervention, the control group had more knowledge than the intervention group.

I'm unclear what point the author was trying to make on p, 9 L. 224-226.  It is confusing

Answer: The sentence was rephrased.

10 L. 260 I assume that "address this issue" should be followed by 'with their parents'

Answer: This information was added in the revised version of the manuscript.

p.10 L. 266-268 doesn't seem to fit.  I think it would help to explain why it is there.

Answer: We agree with the reviewer comment. This sentence was removed from here, since it is not closely related with the discussed issue.

p 10 2nd to last paragraph, L 277-283 is unclear in terms of what point is being made and supported by the literature

Answer: Thank you for the reviewer comment. Discussion section was strengthened and supported by adding more literature data.

There are too many tables in this relatively short paper.  Perhaps some of them can be combined or footnoted.

Answer: This article has 3 tables; the first containing group characteristics and general knowledge on sex education before the intervention; the second is related to the second part of the questionnaire, specifically assessing changes in the participants' general knowledge on sexuality before and after the intervention. The third specifically addresses knowledge on contraceptives before and after the intervention. The reason we did not merge tables 2 and 3 was due to the overall length and organization of the data presented. However, if the reviewer wishes, we can merge them.

This manuscript is a resubmission of an earlier submission. The following is a list of the peer review reports and author responses from that submission.

Round 1

Reviewer 1 Report

The proposed article "Implementation of an Educational Intervention Program on Sexuality and Human Reproduction in Secondary Schools of Huambo, Angola" deals with a very important topic that is seldom appropriately discussed in that part of the globe and I would like to congratulate the Authors on the chosen topic. I have several comment regarding this manuscript, that I hope will improve the overall quality.

Abstract: nicely constructed, however, requires more information on the information. Who carried out these interventions? Teachers in the school or from someone from the outside? How was this intervention implemented in the curriculum?

Introduction: Requires language editing and would benefit for a more streamlined approach. The jump from teenage pregnancy to education seems very clumsy and I suggest the authors structure the introduction to first list the issues in Angola and other neighboring countries and then propose education as one of the possible solutions and support that with the references they already have. 

Methods:What makes this study "quasi-experimental"? What study course did the participating university students study? Medicine? How were the schools selected? What role did the community stake holders and the Church have in the the phase of creating the questionnaire? Who did the lectures? How were these people trained? What was the role of teachers? How were the students chosen? Did all students in the school attend, or only some? How were the students themselves recruited? You also mention talking about psychological maturity. Have you embarked on topics such as premarital sex or same-sex relationships in your lectures?

Results: You have a big difference in the structure of the sample between CG and EG. How did you control for these differences? 

Discussion: somewhat long but I found it informative. 

Overall: language and structural editing would go a long way and help this manuscript improve in quality. 

Author Response

The proposed article "Implementation of an Educational Intervention Program on Sexuality and Human Reproduction in Secondary Schools of Huambo, Angola" deals with a very important topic that is seldom appropriately discussed in that part of the globe and I would like to congratulate the Authors on the chosen topic. I have several comments regarding this manuscript, that I hope will improve the overall quality.

Answer: Thank you for the overall appreciation of our work

Abstract

- nicely constructed, however, requires more information on the information. Who carried out these interventions?

Answer: The interventions were carried out by the principal investigator of the study together with the teachers of moral and civic education and biology of the school selected for the experimental group.

- Teachers in the school or from someone from the outside?

Answer: Professors of the same school

How was this intervention implemented in the curriculum?

Answer: The intervention was performed on non-working days, on Saturday mornings (8 to 10 hours), so as not to interfere with the school calendar.

Introduction: Requires language editing and would benefit for a more streamlined approach. The jump from teenage pregnancy to education seems very clumsy and I suggest the authors structure the introduction to first list the issues in Angola and other neighboring countries and then propose education as one of the possible solutions and support that with the references they already have.

Answer: Introduction was completely restructured

Methods

- What makes this study "quasi-experimental"?

Answer: We consider this study quasi-experimental because it was designed, elaborated and given to the students a study manual containing unknown themes on anatomy, reproduction physiology and contraception, verified during the first survey, and after that an educational intervention was implemented on this subject.

- What study course did the participating university students study? Medicine?

Answer: There were no university students in the two study groups (CG and EG). Only during the preparatory stage, among the students selected for convenience to validate the questionnaire, students from the 1st and 2nd cycle of secondary education (from 6th to 12th grade) and from the 1st year of the Faculty of Medicine were selected. Secondary school teachers and doctors of pediatrics and gynecology and obstetrics were also consulted.

- How were the schools selected?

Answer: Schools were selected by convenience due to logistic and operational reasons.

- What role did the community stake holders and the Church have in the phase of creating the questionnaire?

Answer: Thank you for this pertinent question. Some wanted us to limit the questions to ask and others to increase. In the teaching program, for example, some wanted us to remove Condon from the illustrative materials for teaching about contraception, claiming that it could stimulate anarchic sexual intercourse. After clarification and discussion between the principal investigator and parents and guardians and members of the Council of Christian Churches, it was unanimously accepted to speak openly on all topics.

- Who did the lectures?

Answer: Lectures were given by the principal investigator, accompanied by the school's teachers in subjects related to the teaching program: moral and civic education, biology and the coordinator of extracurricular activities.

- How were these people trained? What was the role of teachers?

Answer: Teachers were present to emphasize their importance in teaching any subject and to better accept pupils and guardians as also to improve their recognition in case of a future implementation of the teaching program on sexuality in schools.

- How were the students chosen?

Answer: This information was included in lines 114-118

- Did all students in the school attend, or only some?

Answer: Only those who agreed to be part of the study.

- How were the students themselves recruited?

Answer: This information was included in lines 114-118

- You also mention talking about psychological maturity. Have you embarked on topics such as premarital sex or same-sex relationships in your lectures?

Answer: Yes. We talk about premarital relationships and the disadvantages of early onset of sex life. In the topics covered, we talked about early sex and its risks, STDs, early pregnancy and clandestine abortions. We did not talk about homosexual relations because of the silence in our country on this subject, although there are already many homosexual individuals.

Results

- You have a big difference in the structure of the sample between CG and EG. How did you control for these differences?

Answer: In fact, the groups are different. In order to overcome this question, we present, for the questionnaire answers, the differences between the groups before and after the intervention (Table 2).

Discussion

- somewhat long but I found it informative.

Answer: We revised the discussion section.

Overall: language and structural editing would go a long way and help this manuscript improve in quality.

Answer: English language was completely revised.

Reviewer 2 Report

As this study demonstrates, there is an urgent need for evaluation and implementation of relationship and sexuality education for adolescents in Angola. This is an interesting study that has the potential to add somewhat to the limited research in the area but the manuscript is not yet at a standard to be suitable for publication in an academic journal.

My biggest concern with this paper is that I don’t think this can be considered an evaluation of a sex education intervention as the questionnaire mainly gathers data on knowledge of human reproduction more in line with biology than sex education. There are only 3 (perhaps 4) questions that I would consider directly related to elements that we would generally see in a sex education intervention and the authors have missed the opportunity to include questions on sexual attitudes and behaviour that might help determine if the intervention had any impact on those variables. While it is of course essential to educate young people on reproductive biology, I would term this ‘biology’ for this and not sex education. Decades of research has provided us with evidence-based sex education interventions designed to impact on teenage pregnancy and STI rates and we now know, that to be effective, these must target multiple psychosocial variables at the individual, interpersonal and environmental levels. However, there is a dearth of research information on adolescent reproductive health in Angola and I applaud the authors for advancing efforts in this area. I do however think, that as well as making the revisions to the paper outlined below, the authors need to be clear that this is an evaluation of an educational intervention about human reproduction and not a sex education intervention.

Title: The title does not adequately describe the study. This is a quasi-experimental evaluation study rather than an ‘Implementation’ study. ‘Intervention’ and ‘program’ describe the same thing so I would use one or other terms and not both. Suggestion = Effects of an educational intervention on Angolan adolescents’ knowledge of human reproduction: A quasi-experimental study

Abstract:

The terms ‘sex education’ or ‘relationship and sexuality education’ are used more appropriate than ‘sexual education’ (in abstract and throughout manuscript), but perhaps the authors need to highlight the lack of education on human biology rather than relationships and sexuality as this is what the questionnaire measures. Study design inaccurately described: The authors appear to misunderstand the study design and have listed several study designs that are not appropriate e.g. I would not consider a 2-month follow-up longitudinal, the experimental and control groups are non-equivalent so cannot be described as ‘paired’ and case-control studies are retrospective (not prospective). I suggest ‘This quasi-experimental study employed a non-equivalent control group, pre-test post-test design’. An international audience will not understand the difference between secondary and high-schools – in the methods section you also mention that primary schools were included? State how many schools were involved Conclusions do not match the aims of the study - the study did not aim to find out about students ‘attitudes’ toward unprotected sex Keywords are inaccurate given points noted above re focus of the study

Introduction:

Grammatical errors throughout Line 43 ‘without losses’ – what does this mean? Line 46 ‘20,000 girls’ is incorrect (much too low), especially since the next line says that 70,000 died ‘by pregnancy’ Please check and update I would also say that they ‘died of pregnancy related complications’ rather than ‘died by pregnancy’ The term Sexually Transmitted Infections (STIs) rather than ‘diseases’ should be used Paragraph from line 55 – more recent prevalence figures should be reported e.g. see UNAIDS report that give 2018 figures (same for next paragraph) Line 73 – what ‘sub systems’? How does sex education ‘prepare an individual for work’? I suggest the authors reference studies that indicate the physical, social and economic impacts of adolescent pregnancy Line 78 is unclear if measures have been taken to allow implementation how is this a delay? A section related to reproduction education (biology) provision would probably be more appropriate for this study

Methods and Materials:

As above re description of study design Primary schools? Is this a mistake? Do you mean post-primary? Line 97: Why did university students and not secondary school students consult on the development of the questionnaire? Do you mean secondary school and university students? Did the students who took part in the questionnaire pilot represent the target population i,e were they 16-year olds? Line 109 – ‘the second part was common’ – what does that mean? If the questionnaires are to be included I think a shorter summary of content would suffice here. One issue that I have is that the questionnaire contains a lot of questions related to the reproductive organs that I would equate more with ‘biology’ than sex education. There are only a small number of questions that actually relate to sexuality and what would be considered relevant to comprehensive sex education. It is unethical and a breach of confidentiality to name the schools that took part in the study – they should be given a pseudonym or called schools 1, 2 and 3 The intervention is not described adequately – I suggest following the TIDieR checklist Did the control group have any intervention or just continue with ‘practice as usual’? Line 120/21 ‘sessions for dissemination and clarification of doubt’ – what does this mean? Line 149 Questionnaires ‘distributed’ rather than ‘applied’ (here and elsewhere) Line 150 to ‘hurt’ the intervention impact? Line 154 How many students were excluded for reasons noted?

Results

Table 1 – What is the difference between ‘married or marital’?; age of first sexual relation is presented twice with different figures (I see small symbols representing males and females but it would e better to state ‘males’ and ‘females’ as there is space to do so; ‘already been pregnant’ and ‘already aborted’ could be reworded to ‘previous pregnancy; and ‘previous abortion’; ‘as far as their sexual relations come’ could be reworded to ‘sexual experience’; ‘Complete sexual relations ‘ could be reworded to ‘sexual intercourse’ or ‘engaged in penetrative sex’ and ‘age of first complete sexual relation’ to ‘age at first sexual intercourse’; Table 1: Given evidence that suggests the importance of gender specific sexual health interventions, it would be useful if all data were present for males, females and all. Table 2 would be easier to use if variable names (short descriptors of the questions) were use in the first column instead of the question number. No reporting of questions on contraceptive use or about concerns around having sex – these would have been most interesting for a study of sex education.

Discussion:

I would have like to have seen a discussion of how these findings relate to other research conducted in Angola (if possible) Line 248: ‘many students revealed they did not know the meaning’. I’m not surprised by this but I am surprised that it did not emerge in your initial consultations with students (perhaps you did not pilot with the target population) Lines 253- 257 beginning ‘This means that sexual relations…’ Be careful of your wording here. It reads like a negative judgment on adolescent sexual behaviour. Engagement in sexual intercourse is a normative behaviour in adolescence (and as you know legal above age 14 in Angola) and comprehensive sex education is directed towards educated decision-making not encouraging abstinence.

Conclusion:

The conclusion should clearly state the main finding of the study at the beginning I don’t see how the findings indicate ‘a careless attitude towards unprotected sexual relations’ – attitudes towards contraception and contraception use were not reported. ‘With this study, effective sexual education has proven to be essential’ – reword There is a need for rigorous development and evaluation of interventions that target multiple individual and environmental level factors

Questionnaires:

Why were females asked in such depth about their menstrual cycle? This doesn’t seem relevant to the study. Would boys remember the age at which they first had an erection?! This happens in early childhood and I don’t see why it is relevant to the study. I understand the questionnaire will have been translated but some of the questions are quite unclear. Qp6A asks for ‘the most correct answer’ but contraception (if the translated term is intended to include condoms as is the case with the English term) equally helps avoid pregnancy and STIs

Author Response

As this study demonstrates, there is an urgent need for evaluation and implementation of relationship and sexuality education for adolescents in Angola. This is an interesting study that has the potential to add somewhat to the limited research in the area but the manuscript is not yet at a standard to be suitable for publication in an academic journal.

Answer: Thank you for the overall appreciation of our work.

My biggest concern with this paper is that I don’t think this can be considered an evaluation of a sex education intervention as the questionnaire mainly gathers data on knowledge of human reproduction more in line with biology than sex education. There are only 3 (perhaps 4) questions that I would consider directly related to elements that we would generally see in a sex education intervention and the authors have missed the opportunity to include questions on sexual attitudes and behavior that might help determine if the intervention had any impact on those variables. While it is of course essential to educate young people on reproductive biology, I would term this ‘biology’ for this and not sex education. Decades of research has provided us with evidence-based sex education interventions designed to impact on teenage pregnancy and STI rates and we now know, that to be effective, these must target multiple psychosocial variables at the individual, interpersonal and environmental levels. However, there is a dearth of research information on adolescent reproductive health in Angola and I applaud the authors for advancing efforts in this area. I do however think, that as well as making the revisions to the paper outlined below, the authors need to be clear that this is an evaluation of an educational intervention about human reproduction and not a sex education intervention.

Answer: We completely agree with the reviewer position. We changed the manuscript focus accordingly.

Title: The title does not adequately describe the study. This is a quasi-experimental evaluation study rather than an ‘Implementation’ study. ‘Intervention’ and ‘program’ describe the same thing so I would use one or other terms and not both. Suggestion = Effects of an educational intervention on Angolan adolescents’ knowledge of human reproduction: A quasi-experimental study.

Answer: We agree with the reviewer suggestion. The title was changed accordingly.

Abstract:

- The terms ‘sex education’ or ‘relationship and sexuality education’ are used more appropriate than ‘sexual education’ (in abstract and throughout manuscript), but perhaps the authors need to highlight the lack of education on human biology rather than relationships and sexuality as this is what the questionnaire measures.

Answer: These aspects were carefully revised throughout the manuscript.

- Study design inaccurately described: The authors appear to misunderstand the study design and have listed several study designs that are not appropriate e.g. I would not consider a 2-month follow-up longitudinal, the experimental and control groups are non-equivalent so cannot be described as ‘paired’ and case-control studies are retrospective (not prospective). I suggest ‘This quasi-experimental study employed a non-equivalent control group, pre-test post-test design’.

Answer: These aspects were included in the revised version of the manuscript.

- An international audience will not understand the difference between secondary and high-schools

Answer: In Angola the education is divided into: Primary education (1stto 5thgrade), secondary 1 cycle (6thto 9thgrade) and secondary 2 cycle (10thto 12thgrade) education. Pre-university education is within the 2nd cycle of secondary education (10thto 12thgrade).

- In the methods section you also mention that primary schools were included?

Answer: The EG had students from both primary and secondary education. Only students from the 8thgrade of the 1stcycle of secondary education were included.

- State how many schools were involved

Answer: In this study, 3 schools were included, 2 public and 1 private.

Conclusions do not match the aims of the study - the study did not aim to find out about students ‘attitudes’ toward unprotected sex

Answer: Conclusion section was restructured.

Keywords are inaccurate given points noted above refocus of the study

Answer: Keywords were updated.

Introduction:

Grammatical errors throughout

Line 43 ‘without losses’ – what does this mean?

Answer: Revised.

Line 46 ‘20,000 girls’ is incorrect (much too low), especially since the next line says that 70,000 died ‘by pregnancy’ Please check and update I would also say that they ‘died of pregnancy related complications’ rather than ‘died by pregnancy’

Answer: Revised

The term Sexually Transmitted Infections (STIs) rather than ‘diseases’ should be used Paragraph from line 55

Answer: This aspect was revised in the entire manuscript.

More recent prevalence figures should be reported e.g. see UNAIDS report that give 2018 figures (same for next paragraph)

Answer: These data were included

Line 73 – what ‘sub systems’?

Answer: Subsystems mean the different levels of education, from primary, secondary (1stand 2ndcycle) and then universities.

How does sex education ‘prepare an individual for work’?

Answer: It was a mistake. This is not sex education, but education in general.

Suggest the authors reference studies that indicate the physical, social and economic impacts of adolescent pregnancy

Answer: Reference 3 was included

Line 78 is unclear if measures have been taken to allow implementation how is this a delay?

Answer: This refers to the non-approval in the Angolan parliament of the basic law and of a specific school program for each level of education in the country. Despite the high rate of teenage pregnancy, unsafe abortions and maternal and infantile deaths, the program and teaching on sex education has not yet been implemented. Indeed, the delay is at the level of legislation and approval of a school education on sexual and reproductive education and contraception.

Methods and Materials:

- As above re description of study design Primary schools? Is this a mistake? Do you mean post-primary?

Answer: Yes

- Line 97: Why did university students and not secondary school students consult on the development of the questionnaire?

Answer: Yes, secondary school’ students from both 1stand 2ndcycle were consulted. We also included some from the 1styear of the university to get an idea if they already had some mastery of these topics. Secondary school teachers and doctors and nurses from pediatrics and gynecology and obstetrics were also consulted.

- Do you mean secondary school and university students? Did the students who took part in the questionnaire pilot represent the target population i,e were they 16-year olds?

Answer: The students chosen in the pilot questionnaire were of the same age group. We just included some students of the 1styear of the university to get an idea if the topics related to this subject were already covered in the universities.

- Line 109 – ‘the second part was common’ – what does that mean?

Answer: In the questionnaire there were a number of personal questions about different gynecological history and sexuality in women and men. In the other part (Part 2), the questions were common to both genders, seeking to know previous knowledge of the anatomy, physiology, contraception and risks of unprotected sexuality.

- If the questionnaires are to be included I think a shorter summary of content would suffice here. One issue that I have is that the questionnaire contains a lot of questions related to the reproductive organs that I would equate more with ‘biology’ than sex education. There are only a small number of questions that actually relate to sexuality and what would be considered relevant to comprehensive sex education. It is unethical and a breach of confidentiality to name the schools that took part in the study – they should be given a pseudonym or called schools 1, 2 and 3 The intervention is not described adequately – I suggestfollowing the TIDieR checklist

Answer: Thank you for the reviewer advice. All aspects were carefully revised.

- Did the control group have any intervention or just continue with ‘practice as usual’?

Answer: There was no intervention. The statistical purpose was to make an intragroup comparison, each student with himself, so paired, before and after, to see if there was any difference between the students with himself before and after, both in EG and CG.

- Line 120/21 ‘sessions for dissemination and clarification of doubt’ – what does this mean?

Answer: It refers to the doubts of parents, guardians and society, and the Council of Christian Churches. Many people in Angola do not talk about sexual issues with their children, some because think their children are still young and should not talk about it anytime soon, while others think that contraceptive use, like Condon cannot be reported, because it may lead to an anarchic use of contraceptive. In the Catholic Church, there is still the concept that it is biblically incorrect to make use of pills, because it would be causing medical abortions. We clarified and exemplified that not using contraceptives is more harmful to the society and cause greater sin among women because it led to unsafe abortions, as abortion is a crime in Angola. These arguments helped to convince all stakeholders.

- Line 149 Questionnaires ‘distributed’ rather than ‘applied’ (here and elsewhere) Line 150 to ‘hurt’ the intervention impact?

Answer: All aspects were revised carefully.

- Line 154 How many students were excluded for reasons noted?

Answer: 59 were excluded. Initially there were 589 students before the intervention; in the post-test, we only had 530. The reasons were for incomplete information, not putting the identification code in the post-test, not attending at least 4 of the 8 lessons of the course.

Results

Table 1

- What is the difference between ‘married or marital’?

Answer: Married are those who are officially married by the civil registry. Those living maritally, are those couples who live together without being officially (legally) married.

- age of first sexual relation is presented twice with different figures (I see small symbols representing males and females but it would e better to state ‘males’ and ‘females’ as there is space to do so; ‘already been pregnant’ and ‘already aborted’ could be reworded to ‘previous pregnancy; and ‘previous abortion’; ‘as far as their sexual relations come’ could be reworded to ‘sexual experience’; ‘Complete sexual relations ‘ could be reworded to ‘sexual intercourse’ or ‘engaged in penetrative sex’ and ‘age of first complete sexual relation’ to ‘age at first sexual intercourse’

Answer: All aspects highlighted were carefully reworded

- Given evidence that suggests the importance of gender specific sexual health interventions, it would be useful if all data were present for males, females and all.

Answer: Really in the initial questionnaire, we had questions about these topics. For example, if "already had full sexual intercourse", "at what age" and "what were the motivations", and if "after consummating the sexual intercourse did you feel any remorse" and "what kind of concern was that?" The vast majority left the question blank and so we had to withdraw it in our results due to lack of answers. Many of questions we wished to study had to be excluded due to the silence of the students, even though it was an anonymous questionnaire.

Table 2 would be easier to use if variable names (short descriptors of the questions) were use in the first column instead of the question number.

Answer: We did not describe due to space reason. The tables are too big and have no space to the numbers.

No reporting of questions on contraceptive use or about concerns around having sex – these would have been most interesting for a study of sex education.

Answer: We agree with the reviewer advice. We just put in the questionnaire what we could ask and get an answer. For example, the type of contraceptive they knew and could mention those who had heard; about who should use contraceptives, we saw that there was clearly a lack of knowledge. Regarding other questions about contraceptives, the students did not answer. The same was true of being pregnant or aborted. They did not answer such questions. We also did not ask anything specific about sexually transmitted infections because we knew it would behave the same way: no answer.

Discussion:

- I would have like to have seen a discussion of how these findings relate to other research conducted in Angola (if possible)

Answer: we really searched for gender studies in Angola and but we did not find any published.

- Line 248: ‘many students revealed they did not know the meaning’. I’m not surprised by this but I am surprised that it did not emerge in your initial consultations with students (perhaps you did not pilot with the target population

Answer: The pilot study was done with 20 students selected for convenience; however, it is possible that being only 20, it was not enough to have a real idea of the problem. But, even in this small number of students, they could not answer both questions. We think it is serious that students about to enter university (11thand 12thgrade and 1styear of the university) fail to take a test on these subjects, while they are all in active sex life or even with children, some of them.

Lines 253- 257 beginning ‘This means that sexual relations…’ Be careful of your wording here. It reads like a negative judgment on adolescent sexual behavior. Engagement in sexual intercourse is a normative behavior in adolescence (and as you know legal above age 14 in Angola) and comprehensive sex education is directed towards educated decision-making not encouraging abstinence.

Answer: We agree with the reviewer suggestion. The paragraph was changed accordingly.

Conclusion:

- The conclusion should clearly state the main finding of the study at the beginning. I don’t see how the findings indicate ‘a careless attitude towards unprotected sexual relations’

Answer: In Table 1, we can see several indicators that led to the above stated conclusion: age of sexual activity onset is lower in younger students belonging to EG: EG (mean age 13 years) vs control group (mean 14 years); regarding remorse or worry about having unprotected sex, in both groups, the majority answered not to fear anything (74.5% in CG vs 25.5% in EG). As for the first sexual vaginal contact, 58.3% of the EG vs 41.7% of the CG had their first full sexual contact before age 18, which reveals the precocity of which young people are having full sexual relations. As for the knowledge and use of contraception, Table 3 shows that most students did not use barrier (male and female condoms, and diaphragm) as well as natural (sexual abstinence) methods, which protects not only from unwanted pregnancies, but also from STI. Anyway, if the reviewer intends, we are open to modify the sentence by the reviewer suggestion.

- attitudes towards contraception and contraception use were not reported.

Answer: Thank you for the reviewer advice. Table 3 summarizes the responses to knowledge and use of the different contraceptive methods available in Angola.

‘With this study, effective sexual education has proven to be essential’ – reword There is a need for rigorous development and evaluation of interventions that target multiple individual and environmental level factors

Answer: We agree with the reviewer suggestion.

Questionnaires:

- Why were females asked in such depth about their menstrual cycle? This doesn’t seem relevant to the study. Would boys remember the age at which they first had an erection?!

Answer: Most of students did not answer these questions, some because did not know what means erection and other ones did not remember the age.

- This happens in early childhood and I don’t see why it is relevant to the study. I understand the questionnaire will have been translated but some of the questions are quite unclear. Qp6A asks for ‘the most correct answer’ but contraception (if the translated term is intended to include condoms as is the case with the English term) equally helps avoid pregnancy and STIs

Answer: Yes, we agree with the reviewer's comment. Looking at questions that the reviewer considered irrelevant, we chose to include those that could provide us as much information as possible. As explained above, most students have not answered some key questions, and in this regard, we have tried to include others that, in addition to provide us more data from students’ knowledge on human biology and reproduction, have also helped us to conclude more accurately on other aspects. Analyzing condom use, we agreed with the reviewer and, in this study, it was considered as having a dual function, contraception and avoid STI.

Round 2

Reviewer 1 Report

I would like to thank the Authors for the changes made. However, in my view there is still a need to both structurally and linguistically edit the paper. 

Furthermore, as you say that only SOME students were reached for the workshops then you have a selection bias as we can hypothesise that those who were more interested in sex education took the course. These students probably come from more open families where these topics are more openly discussed and most probably had previous knowledge that could lead to some data distortion. These limitations should be added. 

Reviewer 2 Report

I appreciate that the authors have taken the time to respond to my comments but, unfortunately, the issues I raised have not been addressed adequately. While the authors have attempted clarification in their responses to my comments, they have not, in most instances, included this information in the article. This will be of no help to other readers! For this reason, I am sorry to say, that I do not recommend this revised version for publication in this journal. Should the authors decide to resubmit the paper I highly recommend that they address the issues I have highlighted.